# Transfersome Encapsulated with the R-carvedilol Enantiomer for Skin Cancer Chemoprevention

**DOI:** 10.3390/nano13050929

**Published:** 2023-03-03

**Authors:** Md Abdullah Shamim, Ayaz Shahid, Pabitra K. Sardar, Steven Yeung, Jeremiah Reyes, Jenny Kim, Cyrus Parsa, Robert Orlando, Jeffrey Wang, Kristen M. Kelly, Frank L. Meyskens, Bradley T. Andresen, Ying Huang

**Affiliations:** 1Department of Pharmaceutical Sciences, College of Pharmacy, Western University of Health Sciences, Pomona, CA 91766, USA; 2College of Osteopathic Medicine of the Pacific, Western University of Health Sciences, Pomona, CA 91766, USA; 3Department of Pathology, Beverly Hospital, Montebello, CA 90640, USA; 4Department of Dermatology, University of California, Irvine, CA 92697, USA; 5Departments of Medicine and Biological Chemistry, Chao Family Comprehensive Cancer Center, University of California, Irvine, CA 92868, USA

**Keywords:** β-blocker, carvedilol, R-carvedilol, ultraviolet, skin cancer, chemoprevention, transfersome, local delivery

## Abstract

The R-carvedilol enantiomer, present in the racemic mixture of the chiral drug carvedilol, does not bind to the β-adrenergic receptors, but exhibits skin cancer preventive activity. For skin delivery, R-carvedilol-loaded transfersomes were prepared using various ratios of drug, lipids, and surfactants, and characterized for particle size, zeta potential, encapsulation efficiency, stability, and morphology. Transfersomes were compared for in vitro drug release and ex vivo skin penetration and retention. Skin irritation was evaluated by viability assay on murine epidermal cells and reconstructed human skin culture. Single-dose and repeated-dose dermal toxicity was determined in SKH-1 hairless mice. Efficacy was evaluated in SKH-1 mice exposed to single or multiple ultraviolet (UV) radiations. Transfersomes released the drug at a slower rate, but significantly increased skin drug permeation and retention compared with the free drug. The transfersome with a drug–lipid–surfactant ratio of 1:3:0.5 (T-RCAR-3) demonstrated the highest skin drug retention and was selected for further studies. T-RCAR-3 at 100 µM did not induce skin irritation in vitro and in vivo. Topical treatment with T-RCAR-3 at 10 µM effectively attenuated acute UV-induced skin inflammation and chronic UV-induced skin carcinogenesis. This study demonstrates feasibility of using R-carvedilol transfersome for preventing UV-induced skin inflammation and cancer.

## 1. Introduction

Skin cancer is one of the most common types of malignancy in the US and globally. One important strategy in controlling skin cancer is avoiding excessive sun exposure and applying sunscreen products. However, up to now, there is inadequate evidence as to whether sunscreen use can reduce the risk of skin cancer. Additionally, sunscreen has its own side effects and limitations [1]. Chemoprevention, which is defined as the use of natural products or pharmacological agents, to inhibit, block, or reverse cancer development, was proposed to reduce skin cancer [2]. There is an increasing interest in chemoprevention for individuals with increased risk for skin cancer. 

The cardiovascular drug carvedilol was reported with skin cancer preventive activity [3,4]. Carvedilol is a β-adrenergic receptor (β-AR) antagonist and an FDA-approved drug used for cardiovascular diseases. Carvedilol has two oral formulations: the immediate-release formulation, which is taken twice a day and the controlled-release formulation, which is taken once a day [5]. Carvedilol is a chiral drug; it is marketed as a racemic mixture consisting of S- and R- enantiomers in a 1:1 ratio. Although S- and R-carvedilol have the same chemical formula, the two enantiomers exhibit distinct pharmacokinetic and pharmacodynamic profiles [6,7,8,9]. Noticeably, S-carvedilol is a potent, competitive antagonist for β-adrenergic receptors, while R-carvedilol is not; it is not considered a β-blocker [10,11]. Our previous studies demonstrate that skin cancer preventive activity of carvedilol is independent of β-AR antagonism [12]. The racemic carvedilol, S-carvedilol, and R-carvedilol similarly prevent ultraviolet (UV) radiation-induced skin DNA damage, reactive oxidative species (ROS) formation, inflammation, and carcinogenesis [13]. Using R-carvedilol can avoid unwanted cardiovascular effects for a chemopreventive agent because they lack β-AR antagonism. In a previous study using an oral dose of 1.6 mg/kg/day, R-carvedilol did not affect heart rate and blood pressure in mice [6]. Thus, the optically pure R-carvedilol enantiomer may be a better candidate for development as a skin cancer chemopreventive agent. 

One limitation in repurposing R-carvedilol for skin cancer prevention is related to its delivery. Carvedilol belongs to the biopharmaceutical classification system (BCS) class II drugs; a highly lipophilic compound with low solubility and poor oral bioavailability [14]. A larger dose and higher dosing frequency are required to achieve the effective concentration. Topical drug delivery shows significant advantages for drugs targeting the skin because it avoids first-pass metabolism and reduces systemic effects [15]. Topical administration is important for skin cancer prevention because the drug will have a greater likelihood of reaching the site of damage and provides a relatively easy method of self-treatment. 

Carvedilol may be encapsulated in surfactant systems or nanocarriers for enhanced dermal targeting [16]. Previously, it was demonstrated that racemic carvedilol can be encapsulated into transfersomal formulations for topical application [17,18]. Transfersome, also named flexible liposome or deformable liposome, is an altered version of conventional liposomes prepared with phospholipids with the addition of surfactants, i.e., edge activators. Transfersomes exhibit an ability to enhance drug penetration into the intercellular lipid matrix by blending with the *stratum corneum* (SC) and modifying the lipid lamellae [19,20,21]. Previous studies showed that transfersome-encapsulated drugs were able to penetrate into deeper layers of skin without systemic absorption [19,21,22]. In contrast, classic liposomes have little value for topical drug delivery because they do not deeply penetrate the skin, but rather remain confined to the SC layer [20].

Therefore, we hypothesized that transfersomes can deliver R-carvedilol into the skin and that R-carvedilol-loaded transfersomes can be developed into a topical formulation for skin cancer prevention. Although a skin targeting transfersome delivery system for the racemic carvedilol was reported [17,18], due to the different stereochemistry of R- and S-carvedilol, it is necessary to examine whether transfersomes can be optimized to effectively deliver the optically pure R-carvedilol. Thus, the goal of the present study is to prepare several R-carvedilol-loaded transfersomal formulations, examine their drug penetration through the skin, deposition into the skin, as well as efficacy and safety. The data presented in this report collectively support a hypothesis that transfersomes are a valid system for R-carvedilol skin delivery. 

## 2. Materials and Methods

### 2.1. Chemicals and Reagents

R-carvedilol was synthesized by Chem-Impex International, Inc. (Wood Dale, IL, USA). The purity was determined by the manufacturer using Chiral HPLC as 98.97%. After the compound was received, the accuracy and purity were confirmed by chiral HPLC using Phenomenex Lux^®^ 5 µm Cellulose-4 LC Column 250 × 4.6 mm (Phenomenex, Torrance, CA, USA). Tween-80, sodium cholate, and polyethylene glycol 400 (PEG 400) were purchased from VWR (Radnor, PA, USA). L-α-phosphatidylcholine (Soy PC or SPC), L-α-phosphatidylcholine hydrogenated (Hydro Egg PC, HEPC), and 1,2-distearoyl-sn-glycero-3-phosphocholine (DSPC) were purchased from Avanti Polar Lipids, Inc. (Alabaster, AL, USA). Carbopol^®^ 934 was purchased from SERVA Electrophoresis GmbH (Heidelberg, Germany). Triethanolamine (TEA) was purchased from Sigma-Aldrich (St. Louis, MO, USA).

### 2.2. Preparation of R-carvedilol-Loaded Transfersomes and Carbopol Gel

Transfersomes were prepared by a thin film hydration method as described previously [18]. In brief, the lipids, surfactants and R-carvedilol (5 mg) were dissolved in chloroform:methanol (2:1, *v*/*v*). To form a thin film, the organic solvent was gradually evaporated under reduced pressure in a rotary evaporator at 45 °C for 30 min. Next, the thin film was hydrated in 10 mL PBS (pH 7.4) at 51 °C. Then, the formulation was sonicated in a water bath for 5 or 30 min before passing through a 100 nm pore size membrane (Avanti Polar Lipids, Alabaster, AL, USA) through an extruder (Liposofast LF-50, Avestin, Ottawa, ON, Canada) to reduce the particle size and obtain stable transfersomes. The plain transfersome (PT), used as no drug control, was prepared in the same way, except that no drug was added. Carbopol gel was prepared because it was previously reported in mouse studies that it increases skin retention [17]. The transfersomal formulations were mixed with 0.5% Carbopol^®^ 934 and triethanolamine (TEA) (1:1.5, *w*/*w*) and then vortexed until a clear gel was formed. 

### 2.3. Determination of Particle Size, Zeta Potential and Encapsulation Efficiency

Particle sizes and polydispersity index (PDI) were determined using the Nanobrook Omni particle sizer (Brookhaven Instruments Corporation, Holtsville, NY, USA). Zeta potential was determined using Malvern zeta-sizer (Malvern Panalytical, Malvern, UK). The encapsulation efficiency (EE) was determined according to reported methods [18]. In brief, the transfersomes were centrifuged inside the 30,000-dalton cutoff Nanosep^®^ tubes (Pall Life Sciences, Ann Arbor, MI, USA) at 14,000 rpm for 1 h at 4 °C. The drug level in the filtrate, which represents the free drug was analyzed via HPLC. To determine the total drug concentration, the transfersome suspension (10 μL) was mixed with 990 μL of methanol and vortexed for 1 h to disrupt the transfersomes. The encapsulation efficiency was calculated by the following formula.
$$\mathrm{Encapsulation}\;\mathrm{efficiency}\;\left(\%\right)=\frac{\left(total\;\mathrm{drug}\;concentration-free\;\mathrm{drug}\;concentration\right)}{total\;\mathrm{drug}\;concentration}\times 100$$

### 2.4. In Vitro Drug Release Analysis

The in vitro drug release was analyzed using a Pur-A-Lyzer Mini Dialysis Kit with 3.5 kDa as the molecular weight cut off (MWCO) (Sigma-Aldrich, St. Louis, MO, USA). The release was conducted in a shaking incubator at 100 rpm and 37 °C. The transfersomes containing 0.1 mg R-carvedilol or the same amount of free drug dissolved in PEG 400 (volume 2.5 mL) were added into the dialysis tubes. PBS was used as the release medium and the tubes were immersed in 100 mL of PBS (pH 7.4). At various time intervals (0.5, 1, 2, 3, 4, 6, 8, and 24 h), 1 mL samples were withdrawn from the release medium and replaced with 1 mL of fresh PBS. The samples were analyzed via HPLC.

### 2.5. Ex Vivo Skin Permeation Analysis

The ex vivo skin permeation and retention studies were performed using the Franz diffusion system (Crown Glass Company, Somerville, NJ, USA) (surface area of 1.13 cm^2^), loaded with porcine ear skin (Sierra for Medical Science, Whittier, CA, USA). The methods were previously reported [17,18]. In brief, the receiver compartment was filled with 5.5 mL of 40% *v*/*v* PEG 400 in PBS (pH 7.4) or 4% bovine serum albumin (BSA) in PBS and maintained at 37 °C under magnetic stirring. PEG 400 or BSA was added to increase the solubility of permeated R-carvedilol. T-RCAR transfersomes or the free drug dissolved in PEG 400 containing 4 µg R-carvedilol (200 mL) were applied onto the porcine skin in the donor compartment. At the time intervals of 0.5, 1, 2, 3, 4, 6, 20, and 24 h, 0.2 mL solvent in the receiver compartment was collected and replaced with 0.2 mL solvent. 

### 2.6. Transmission Electron Microscopic (TEM) Analysis

TEM is a visualization tool for nanoparticles and was used to obtain a quantitative measure of particle size and size distribution [23]. Vesicles that are suspended in PBS were examined by TEM with the negative staining method at Fortis Life Sciences/nanoComposix (San Diego, CA, USA). In brief, samples were prepared for imaging by drying the nanoparticles on a copper grid coated with a thin layer of carbon. Images were obtained by the use of a JEOL 1010 transmission electron microscope (JEOL USA, Peabody, MA, USA), operating at an accelerating voltage of 100 keV and an AMT XR41-B 4-megapixel (2048 × 2048) bottom mount camera. The camera’s finite conjugate optical coupler provides high resolution and flat focus with less than 0.1% distortion for magnifications as high as 150,000×.

### 2.7. HPLC Analysis

After R-carvedilol was received, chiral HPLC analysis was used to determine the purity and accuracy of R-Carvedilol, with Phenomenex Lux^®^ 5 µm Cellulose-4 LC Column 250 × 4.6 mm (Part No. 00G-4491-E0, Serial Number H21-389311, Batch No. 5599-0063) (Phenomenex, Torrance, CA, USA). The drug level was detected using an Agilent 1260 HPLC system (Agilent Technologies Inc., Santa Clara, CA, USA), equipped with a quaternary pump (G1311B), an autosampler (G7129A), an automatic thermostatic column compartment, a diode array detector (DAD) detector (G1315D), and a computer with Agilent OpenLAB CDS Chemstation Edition for LC&LC/MS Systems (Rev C.01.07). The drug was separated on a BDS Hypersil C18 reverse-phase column (2.1 × 150 mm; 2.4 um) (Thermo Scientific, Waltham, MA, USA) coupled with a C18 guard column (10 mm × 2.1, 3 um) (Thermo Scientific). The mobile phase was a mixture of acetonitrile:buffer (38:62); the buffer was made of 20 mM ammonium acetate, 0.1% triethylamine, and adjusted to pH 4.5 with phosphoric acid. The flow rate was 0.2 mL/min. R-carvedilol was analyzed at the wavelength of 240 nm. Propranolol (1 µg/mL) was used as an internal standard. 

### 2.8. Cell Culture and MTT Assay (2D)

The in vitro cytotoxicity was evaluated in the monolayer culture of the non-tumorous murine epidermal cell line JB6 CI 41-5a (JB6 P+) (ATCC, Manassas, VA, USA) using MTT (3-[4,5-dimethylthiazol-2-yl]-2,5 diphenyl tetrazolium bromide) assay, because this cell line was shown as a commonly used model to predict in vivo skin toxicity of chemicals [23]. JB6 P+ cells were maintained in Eagle’s minimum essential medium (EMEM) containing 4% heat-inactivated fetal bovine serum and 1% penicillin/streptomycin. The cells were seeded in a 96-well plate at a density of 1 × 10^4^ cells/100 μL medium/well and incubated overnight or until 70–80% confluence. The cells were treated with T-RCAR-3, free R-carvedilol, or plain transforsome (PT). The drug concentrations were 0.1, 1, 10, and 100 µM. The PT was diluted the same way as the T-RCAR. MTT assay was conducted 48 h after incubation. The stock solution (5 mg/mL) of thiazolyl blue tetrazolium bromide (MTT; Sigma, M2128) was prepared in DPBS (pH 7.4) and filtered to remove crystals and to sterilize. The MTT solution was added to each well in an amount equal to 10% of the cell culture volume and the plate was incubated at 37 °C for 4 h. Isopropanol with 0.1 N HCl (100 µL) was added to each well to replace the media before using a spectrophotometer at 570 nm wavelength to read the optic density of the formazan salt produced (reference 630 nm).

### 2.9. EpiDerm Skin Irritation Test (3D)

The in vitro skin irritation test for T-RCAR was conducted according to the protocol developed by MatTek Corporation (Ashland, MA, USA) of “EPI-200-STI”, a reconstructed human epidermal model EpiDerm [24]. The EpiDerm skin culture was purchased from MatTek. In brief, eight treatment groups including the negative and positive controls were established on the EpiDerm culture (sample size n = 3). The negative control was DPBS and the positive control was 5% SDS. The test groups consisted of the plain transfersome (same dilution as the 100 μM of T-RCAR), T-RCAR-3 at three doses (10 μM, 20 μM, and 100 μM), PEG 400 as the vehicle for free drug, and 100 μM of a R-carvedilol-free drug dissolved in PEG 400. The EpiDerm culture was exposed to test agents for 60 min and incubated at 37 °C without treatments for 2 days. Then a MTT assay was used to determine cell viability and irritation. After, the same procedure was conducted as the 2D MTT assay. A 96-well plate was used to read the optic density of the formazan salt produced in the Epiderm samples that included two of each treatment group. Then the mean was taken.

### 2.10. Acute and Repeat Dose Dermal Toxicity Study in Mice

All animal studies were carried out under the recommendations and guidelines established by the Western University of Health Sciences’ Institutional Animal Care and Use Committee, which approved these studies. Mice had access to water and food ad libitum and housed on a 12 h light/dark cycle with 35% humidity. All SKH-1 hairless mice were obtained from our internal breeding protocol with breeding pairs purchased from Charles River (Wilmington, MA, USA).

The acute dermal toxicity study (single dose testing) was carried out in accordance with the OECD guidelines No. 402 (2017) with modification. In brief, six female adult healthy SKH-1 hairless mice (6~8 weeks old) were randomly divided into two groups (control and T-RCAR treatment). Although 10 µM of R-carvedilol was effective in our previous studies [13], the toxicity study started at a higher dose, 100 µM in 0.5% Carbopol gel (200 µL). According to our previous study, the gel form and suspension form of transfersomes of carvedilol showed the same skin permeation profile [17]. On day 0, a single dose of T-RCAR gel was applied topically with a uniform distribution over the back area of ~6 cm^2^. Body weight was measured on days 0, 1, 2, 3, and 14. The skin phenotypes were assessed using the multi-probe adapter system (Courage and Khazaka Electronic GmbH, Cologne, Germany), a non-invasive method commonly used to evaluate skin irritation and barrier function, which includes other methods used to evaluate skin dryness, erythema, and trans-epidermal water loss (TEWL). This was conducted at days 0, 1, 2, and 3. 

The repeated dose acute dermal toxicity study was carried out in accordance with the OECD guidelines No. 410 (1981) with modification. In brief, adult healthy mice (6~8 weeks old, females) were used. Animals were randomized into three groups: negative control, plain transfersome (PT), and T-RCAR. Four animals were used as negative control without any treatment. Six animals were used in the PT and in the T-RCAR group, which were then treated with T-RCAR containing 100 μM (200 μL) of drug or the plain transfersomes of the same volume. This was applied topically once every day for 21 days corresponding to the two groups. The treatment started on day 0. On days 0, 1, 2, 3, 6, and 21, the body weight and skin parameters were measured. Animals were euthanized at the end of the study, 24 h after the last dose. Major organs including skin, liver, kidneys, lung, heart, and spleen were excised and stored in formalin for histological evaluation. 

### 2.11. UV-Induced Acute Skin Inflammation

Female SKH-1 hairless mice, seven-eight weeks of age, were randomly divided into four groups (n = 3~4): (1) UV + PEG 400 vehicle, (2) UV + T-RCAR-3 gel containing 10 μM R-carvedilol, (3) UV + free R-carvedilol 10 μM in PEG 400 solution, and (4) UV + plain transfersomes gel. The UV lamps used in these studies were previously described [17]. Topical treatment of drug or vehicle was given on day −2, −1, and day 0 immediately after a single dose of UV radiation (336 mJ/cm^2^). For all topical treatments, a 200 μL volume of test agents were applied to the back of the mouse. The area of treatment for each mouse was approximately 6 cm^2^ of back skin. During the UV exposure, mice roamed freely in acrylic cages on a rotating platform, ensuring consistent and equal dorsal distribution of UV irradiation. Six hours after UV exposure, all mice were euthanized. Whole skin samples were dissected for RNA isolation. 

### 2.12. RNA Isolation and qPCR Analysis

Total RNA was isolated from whole skin tissue using the RNeasy Mini Kit (Qiagen, Germantown, MD, USA). cDNA was synthesized with the High-Capacity cDNA Reverse Transcriptase Kit (Thermo Fisher). cDNA and SYBR Green Supermix (Thermo Fisher) were mixed with primers for mouse IL-6 gene and β-actin (the primer sequences are available upon request). qPCR was performed on a CFX96 real-time thermal cycler detection system (Bio-rad, Hercules, CA, USA) and analyzed with the 2^−ΔΔct^ with β-actin as the normalization control.

### 2.13. Chronic UV-Induced Skin Tumorigenesis

Seven-week-old female SKH-1 mice were randomly divided into four groups (n = 10): (1) UV only control, (2) UV-exposed followed by plain transfersome (PT) gel treatment, (3) UV-exposed followed by T-CAR gel treatment (10 µM), and (4) UV-exposed followed by the free drug R-carvedilol. The volume for topical treatment was 300 μL. Mice were pretreated with drugs three times a week for two weeks before starting UV exposure. The mice were then irradiated with gradually increasing levels of UV three times a week for 25 weeks with an initial dose of 50 mJ/cm^2^. The UV was increased each week by 25 mJ/cm^2^ to 150 mJ/cm^2^, which was continued for the duration of the experiment. The drug was applied topically immediately after UV radiation. Tumors of at least 1 mm in diameter were counted and measured with a caliper weekly. The tumor volume was calculated according to the formula: (width)^2^ × length/2. 

### 2.14. Statistical Analysis

All the data were expressed as a mean ± standard deviation (SD) or standard error (SE) unless stated otherwise. In histograms, all data are shown with a line representing the group mean. All plots were made using GraphPad Prism version 9.2.0 (La Jolla, CA, USA). Statistical analysis of the data was conducted in Prism for one-way ANOVAs, and all other statistical analysis was conducted using NCSS 2019 Statistical Software (Kaysville, UT, USA). The specific statistical tests are detailed in the text and figure legends. For all statistical analyses, means were indicated to be statistically different when *p* < 0.05.

## 3. Results

### 3.1. Transfersome Preparation, Characterization, and Selection

Pilot transfersomes were prepared using three different lipids, including SPC, DSPC, and HEPC, with a constant drug:lipid:surfactant ratio of 1:3:0.5 using Tween 80 as the surfactant. Each transfersome was loaded with 5 mg R-carvedilol and each thin film was hydrated in 10 mL PBS. Only SPC-based transfersomes successfully formed stable nanoparticle suspensions, while formulations prepared with DSPC and HEPC precipitated immediately after the thin films were hydrated in PBS. This is consistent with previous reports that stated using these lipids (DSPC and HEPC) failed to produce carvedilol-loaded transfersomes [18]. Therefore, we decided to use SPC to prepare R-carvedilol-loaded transfersomes. 

To determine the effects of the presence or absence and the concentration of surfactants, transfersomes were made with Tween 80 (nonionic surfactant) or sodium cholate (ionic surfactant), with various drug:lipid:surfactant ratios. These transfersomes demonstrated comparable particle size, polydispersity index (PDI), and zeta potential. EE was higher than 80% (Table 1). The transfersomes made with Tween 80 showed slightly higher penetration across the porcine ear skin (the difference was insignificant, *p* > 0.05) (Table 1). Since transfersomes for carvedilol made with Tween 80 were effective without systemic absorption [17], we decided to focus formulation development on Tween 80-containing transfersomes. 

Transfersomes made with R-carvedilol:SPC:Tween 80 in ratios of 1:3:0, 1:3:0.25, and 1:3:0.5 were named as T-RCAR-1, T-RCAR-2, and T-RCAR-3, respectively. Three batches for each formulation were prepared. Table 2 shows the particle size, PDI, zeta potential, and EE data for all the nine batches of transfersomes. These parameters were similar across different transfersomes and batches, indicating a high degree of reproducibility. It is noticeable that the zeta potential for the newer batches (Table 2) was higher than that for the pilot formulations (Table 1). One possibility is that when we prepared the pilot formulations in Table 1, the suspension was sonicated for 5 min. Later, when we repeated the formulation preparation to make additional batches we sonicated the formulations for 30 min, because we found that longer sonication avoided the formation of clusters and increased the stability.

### 3.2. In Vitro Drug Release Testing for R-carvedilol-Loaded Transfersomes (T-RCARs) 

The in vitro drug release profiles of the free drug (0.1 mg R-carvedilol dissolved in PEG 400 solution) and three different transfersomal formulations containing the same amount of the drug were compared using a dialysis membrane method in a release medium of pH 7.4 PBS, mimicking the physiological pH. As stated above, the difference of the three T-RCARs is the content of Tween 80, with T-RCAR-1 containing 0%, T-RCAR-2 containing 0.0125%, and T-RCAR-3 containing 0.025% Tween 80. Compared with the free drug, the R-carvedilol-loaded transfersomes showed a slower drug release profile (Figure 1). At 24 h, the free drug solution achieved an equilibrium between the inner and outer compartments, while the drug release from T-RCARs only achieved 30~60% of the total drug amount. Among the three transfersomes, T-RCAR-3, which contained the highest concentration of Tween 80, showed higher drug release than the other transfersomes. The in vitro drug release profiles for T-RCAR-3 with drug:SPC:Tween 80 ratio of 1:3:0.5 in the present study were similar to the carvedilol-loaded transfersome with the same drug:SPC:Tween 80 ratio characterized in previous studies [17]. These data indicate that the transfersomes encapsulate the drug and that the amount of Tween 80 affects in vitro drug release.

### 3.3. Ex Vivo Drug Penetration and Skin Retention of T-RCARs

A key parameter for topical delivery is the skin permeation of the drug [16,25]. The Franz diffusion cell system loaded with porcine ear skin was used to determine the skin permeation kinetics for the three T-RCAR formulations in comparison with free drug dissolved in PEG 400. The Franz diffusion cell is a commonly used method for predicting human skin permeability [26]; 4000 ng total R-carvedilol in each formation was applied to the *stratum corneum* side of the skin in the donor chamber. Due to the poor solubility of R-carvedilol in PBS, PBS containing 40% PEG 400 was used as the release media in the receptor chamber. The cumulative drug levels that permeated the skin at various time points are shown in Figure 2. For all formulations, the drug was not detectable within the first 6 h. However, 20 h later, the drug that penetrated through the skin using T-RCARs was much higher than the free drug (Figure 2A). Twenty-four hours after drug loading, only 2.9 ± 3.2% free drug permeated the skin, while 24~29% drug permeation was detected when using T-RCARs. However, the three transfersomes did not show significant difference in terms of drug penetration. After 24 h of drug incubation, the skin was collected, and a tape-stripping technique was performed to analyze the drug deposit in the stripped skin (epidermis and dermis) (Figure 2B). Although the skin penetration data for the three T-RCARs show no difference in Figure 2A, T-RCAR-3 displayed statistically greater skin drug retention than the free drug and T-RCAR-2 (Figure 2B). 

To confirm the ex vivo data by the use of physiologically relevant release media, PBS containing 4% bovine serum albumin (BSA) [27] in the receptor chamber was used to replace PBS containing 40% PEG 400. The percentage of cumulative R-carvedilol permeated at various time points is shown in Figure 2C. Experimental setting using BSA in the receptor chamber showed the same trend. After 24 h, the skin was collected, and a tape-stripping technique was performed to analyze the R-carvedilol in stripped skin. Similar to the PEG 400 receptor fluid shown in Figure 2B, T-RCAR-3 showed the highest skin retention although no statistical difference was detected among the four groups (Figure 2D). 

### 3.4. Stability of T-RCAR Formulations

Based on the ex vivo skin penetration and retention data where T-RCAR-3 displayed the highest skin retention (Figure 2), T-RCAR-3 was selected for a stability test. Stability is an important parameter for formulation development because this data will dictate the shelf life of a product. After being stored at 4 °C for up to 4 weeks (Table 3), three independent batches of T-RCAR-3 were characterized for particle size, PDI, and EE to confirm that T-RCAR-3 can remain an intact nanoparticle for an extended time. All parameters examined were consistently stable throughout the assay period. 

Although the stability test was only planned for T-RCAR-3 for 5 weeks, the formulations were placed in a refrigerator at 4 °C for longer storage. The three formulations started to show some visible difference. As shown in Figure 3A, T-RCAR-1 formed a precipitate after one-year storage, whereas T-RCAR-2 and T-RCAR-3 remained clear solutions. The particle size data indicate that T-RCAR-1 and T-RCAR-2 exhibited increased particle sizes. The average size and PDI for T-RCAR-1 were 141.27 ± 1.34 and 0.22 ± 0.0006, respectively, and the average size and PDI for T-RCAR-2 were 138.64 ± 1.59 and 0.15 + 0.02, respectively. This is possibly due to fusion or aggregation. However, T-RCAR-3 remained the smallest in average size (118.53 ± 0.76) and PDI (0.09 ± 0.03) (Figure 3B), the same size as a year ago (Table 3). The long-term stability data suggest that T-RCAR-3 is more stable than the other two formulations. This data suggest that the presence of Tween 80 plays an important role in enhancing the formulation stability, since T-RCAR-1 does not contain Tween 80, while T-RCAR-3 contains the highest amount of Tween 80.

### 3.5. Morphological Study of T-RCAR-3

TEM analysis was used to examine the morphology of the selected transfersomal formulation, T-RCAR-3. Based on the representative negative staining TEM images (Figure 4A), circular unilamellar lipid vesicles were observed. The size distribution showed that more than 70% of the particles were smaller than 100 nm in size (Figure 4B). Therefore, the TEM image analysis confirmed the successful formation of the transfersome particles.

### 3.6. In Vitro Toxicity and Skin Irritation Study of T-RCAR-3

To determine whether T-RCAR-3 has any possibility of inducing skin toxicity or irritation, we conducted cytotoxicity studies on a monolayer culture (2D) of murine epidermal cells JB6 P+ using the MTT assay. The cells were treated with various concentrations of T-RCAR-3, the free drug R-carvedilol at various concentrations (0.1, 1, 10, and 100 µM), or the plain transfersomes of the same dilutions (n = 3 per treatment group). After 48 h of incubation, at concentrations of 10 µM or below, there was slightly reduced viability observed for T-RCAR-3 (Figure 5A), but the difference was not dose-dependent, and statistically insignificant. However, at concentrations of 100 µM or higher, no viable cells were observed for all three test agents. The observed effects may be attributed to both the drug and the transfersomal vesicle since T-RCAR-3, free drug, and plain transfersomes induced the same cytotoxicity on MTT assay. 

Next, the reconstructed human epidermal model EpiDerm (EPI-200-SIT) model was used to examine the irritation potential of topical T-RCAR, since the monolayer culture did not have the outermost layer of the skin (*stratum corneum*). The EPI-200-SIT was validated for predicting skin irritation potential and for replacing in vivo acute skin irritation test in rabbits [24]. According to MTT data on 2D culture (Figure 5A), doses at 10 µM did not produce any cytotoxicity, so doses at or lower than 10 μM were not examined in the 3D model. MTT assay was used to quantify the damage caused by testing agents. A reduction in MTT reading more than 50% indicated skin irritation in accordance to the manufacturer’s protocol. As seen in Figure 5B, SDS (5% in H_2_O) solution, the positive control, resulted in 4.86 ± 0.28% of viability of the negative control (treated with sterile DPBS) (*p* < 0.0001). However, none of the doses of testing agents significantly affected viability. All the concentrations examined for T-RCAR-3 (10, 20, and 100 µM) showed similar viability as the negative control (*p* > 0.05). This result indicates that 100 μM T-RCAR, which is 10 times higher than an effective dose (10 μM), did not cause skin irritation on EpiDerm. 

### 3.7. In Vivo Dermal Toxicity and Irritation Study of T-RCAR-3

R-carvedilol is present in an FDA-approved drug with low risk of systemic toxicity. In addition to mortality and body weight data, we measured several skin parameters to evaluate potential skin irritation. The acute dermal toxicity study initiated from 100 μM, but in most previous efficacy studies, the effective dose was 10 μM [13]. Single-dose topical treatment of T-RCAR at 100 μM in Carbopol gel in mice (n = 3) did not cause mortality, body weight loss, or abnormal clinical signs. Therefore, a repeated-dose study was performed on new groups of mice treated with 100 μM T-RCAR gel (n = 6) or plain transfersome (PT) gel (n = 6) once every day for 21 days in comparison with mice without any treatment (n = 4). The mice treated with PT or T-RCAR did not show significant difference in body weight in all time points examined (*p* = 0.63) (Figure 6A), which indicated a lack of systemic toxicity. The cutaneous hydration levels measured by a corneometer displayed variability over time in all groups, but no statistical differences were observed (*p* = 0.34) (Figure 6B). The transdermal water loss (TEWL) did not show any statistically significant changes in all time points examined (*p* = 0.79) (Figure 6C). However, there was a statistically significant change in the erythema index (*p* = 0.02) (Figure 6D), due to a slightly increased erythema in the negative control groups. These mice did not display any visible erythema or obvious clinical abnormalities. H&E staining of skin tissues dissected from mice that were treated with T-RCAR-3 demonstrated normal structure and morphology comparable to controls (Figure 6E). Although these data demonstrate the safety of T-RCAR, the potential effects of T-RCAR gel on cutaneous hydration measured by the corneometer should be verified by other approaches, such as optical imaging methods [28].

### 3.8. Effects of T-RCAR-3 on UV-Induced Skin Inflammation and Carcinogenesis in Mice

We exposed the SKH-1 mice to a single UV radiation dose to evaluate the efficacy of T-RCAR-3 on UV-induced skin inflammation. Previous studies showed that skin expression of inflammatory marker IL-6 was upregulated after six hours of UV exposure [13]. Topical drug treatments were started on day 2, day 1, (pretreatment) and day 0 immediately after single dose UV radiation (336 mJ/cm^2^) to evaluate the efficacy of T-RCAR-3. For topical treatment, T-RCAR-3 (10 μM drug) and plain transfersome were applied as Carbopol gel, while free drug R-carvedilol (10 μM) was dissolved in PEG 400. The experiment was terminated 6 h after exposure to UV. As seen from Figure 7A, both T-RCAR-3 and free drug showed a significant reduction in UV-induced mRNA overexpression of IL-6 in comparison with the vehicle control treated with PEG 400. These results are consistent with the ex vivo data, in which all transfersomes and free drug showed certain degrees of skin retention (Figure 2). Although this assay could not distinguish between T-RCAR-3 and free drug dissolved in PEG, it confirms the anti-inflammation efficacy of both. 

Next, we applied the chronic UV-induced skin carcinogenesis protocol to evaluate the cancer preventive effects of T-RCAR-3. In previous chronic UV studies based on the same protocol, R-carvedilol dissolved in acetone (10 µM) showed skin cancer preventive effects [13]. Although acetone as skin penetration enhancer works in a preclinical setting, using acetone on human skin for long periods of time can lead to skin barrier disruption and dermatitis [29]. In this present study, we examined the cancer preventive effects of T-RCAR-3 containing 10 µM R-carvedilol in comparison with the same dose of free R-carvedilol dissolved in acetone as a positive control. T-RCAR-3 statistically decreased total tumor multiplicity (Figure 7B) and tumor volume (Figure 7C) to a similar degree as free drug. However, there were no statistically significant differences in tumor growth rates when comparing R-carvedilol with acetone control (Figure 7D). T-RCAR-3 showed a slower trend of tumor growth compared with PT (Figure 7D), although the difference was not significant. Representative mouse photos are shown in Figure 8 with a visible difference between UV only controls and T-RCAR-3-treated animals. 

## 4. Discussion

The present study developed R-carvedilol-loaded transfersomes, namely “T-RCAR” and characterized these formulations in vitro and in vivo. This study produced essential data to support our hypothesis that T-RCAR formulations could be effective and safe. Several R-carvedilol-uploaded transfersomes were prepared and characterized. Firstly, three phospholipids were used: SPC, DSPC, and HEPC. While DSPC and HEPC failed to make stable transfersomes, SPC-made transfersomes were most stable and used throughout the formulation development. Secondly, two different surfactants were used, Tween-80 and sodium cholate. Both surfactants successfully resulted in stable transfersomes (Table 1). According to a pilot pig ear skin permeation study (Table 1), Tween-80 was selected for development of transfersomes. Thirdly, three transfersomes with different drug:lipid:surfactant ratios, named T-RCAR-1, T-RCAR-2, and T-RCAR-3, were compared in terms of stability, in vitro drug release, and ex vivo skin drug retention (Table 2 and Table 3) (Figure 1, Figure 2 and Figure 3). T-RCAR-3, which contains the highest level of Tween-80 among the three transfersomes, was selected based on the ex vivo skin retention data (Figure 2). For the ex vivo skin permeation and retention study, two different types of fluids were used in the receptor chamber. Since R-carvedilol is a lipophilic compound, using PBS alone may result in lower permeability of R-carvedilol. The first fluid we used was PBS containing 40% PEG 400. PEG 400 was added to the receptor chamber to increase the solubility of R-carvedilol. The same experiment was conducted using PBS containing 4% BSA since PEG 400 is not naturally present in the body. According to the literature report [29], the BSA-containing fluid can better predict transdermal drug delivery for lipophilic compounds. Both fluids consistently showed that T-RCAR-3 had the highest skin retention. Although we do not yet understand the mechanism, Tween 80 level appeared to correlate with stability, in vitro release, and skin retention. Furthermore, the ex vivo skin permeation and retention study was conducted using the Franz diffusion cell system loaded with porcine ear skin. Future studies should confirm this finding on excised human skin.

Although most studies (Figure 4, Figure 5, Figure 6, Figure 7 and Figure 8) were conducted only for selected T-RCAR-3, other transfersomes with different content of Tween 80 may be also effective. However, T-RCAR-1, which does not contain any Tween 80 should not be considered, since after long-term storage, a precipitation was observed (Figure 3). The one-year stability data for the three T-RCAR formulations, although interesting, are limited due to small sample sizes. The long-term stability studies were not planned initially, but focused on T-RCAR-3 for a 5-week stability study, which is shown in Table 3. Therefore, future studies should be directed to verify the long-term stability of these formulations in different storage conditions. In fact, based on the definition of transfersomes, T-RCAR-1 is not classified as a transferosome as it does not contain surfactant but is a conventional liposome. T-RCAR-2 and T-RCAR-3 performed rather similarly in the comparison studies (Figure 1, Figure 2 and Figure 3); however, T-RCAR-3 matches previously reported carvedilol transfersomal formulations [17] and performed marginally better in the comparisons. Therefore, T-RCAR-3 was selected for further testing. 

The EpiDerm model is a physiologically relevant skin vitro skin models consisting of epidermis and dermis. Although the irritating results from EpiDerm are accepted, due to the robust barrier properties of the EpiDerm model, the non-irritating results may require further verification by other methods [30]. The in vitro and in vivo skin irritation studies used a higher dose than the effective dose in previous mouse cancer prevention studies [13]: 100 μM, which is 10 times higher than the effective dose, showed no irritation on the 3D human skin construct and in SKH-1 mice (Figure 5 and Figure 6). Further efficacy studies should be conducted to evaluate whether doses as high as 100 μM can be more effective. Further efficacy studies will also need to evaluate whether doses lower than 10 μM can be effective. The future goal is to identify the minimally effective dose for R-carvedilol transfersomes, which is essential information for future clinical drug development. 

The current study did not provide comprehensive mechanistic data for the R-carvedilol transfersomes. R-carvedilol and the racemic carvedilol, which contains R- and S-carvedilol, attenuated UV radiation-medicated skin lesions via multiple mechanisms of action. In a previous report [13], several assays were used to determine the effects of the free drug R-carvedilol on UV-induced oxidative stress, inflammation, and DNA damage, in vitro and in vivo. In the current study, IL-6 expression was used as the representative proinflammatory biomarker (Figure 7). Future work should investigate the potential mechanisms of action for R-carvedilol’s skin cancer preventive activity. 

Although carvedilol-loaded transfersomes were effective and safe in principle [17], R-carvedilol-loaded transferosomes were preferred over carvedilol-loaded transferosomes for cancer prevention. The reason is that carvedilol is a highly potent β-blocker, with an IC_50_ at nanomolar ranges [31]; skin-targeted formulations may still be absorbed into the systemic circulation, particularly if patients have skin damage due to sunburn or another wound. Additionally, a topical product that is safe if swallowed increases the overall safety profile of the product. Due to the fact that R-carvedilol can effectively prevent UV-induced skin damage at much lower doses (0.1 μM) [13], it is feasible to develop an over-the-counter topical formulation. The potential clinical applications for such a topical formulation include preventive treatments for a range of skin lesions associated with exposure to UV radiation, including sunburn, actinic keratoses (pre-cancerous skin lesions), and skin cancer, including both non-melanoma and melanoma skin cancer. A T-RCAR formulation could be applied by any individual before sunlight exposure or after a sunburn. However, additional preclinical studies are needed before clinical trials can be started. Specifically, all the efficacy studies presented in the present study have a pre-treatment paradigm (Figure 7), which needs further evaluation as pre-treatment may not be feasible for all people. 

## 5. Conclusions

This study provides preclinical evidence that R-carvedilol-loaded transfersomes can effectively prevent UV-induced skin cancer without any major adverse effects. Although the transfersome-based topical drug delivery system is proven to be stable, safe, and effective in the preclinical experimental setting, further work should identify the optimal dose of topical R-carvedilol in the chemoprevention of skin cancer. 

## 6. Patents

US Utility Patent Application (serial no. 17/676,684) was filed on 21 February 2022 for the application of R-carvedilol in cancer chemoprevention.

## Figures and Tables

**Figure 1 nanomaterials-13-00929-f001:**
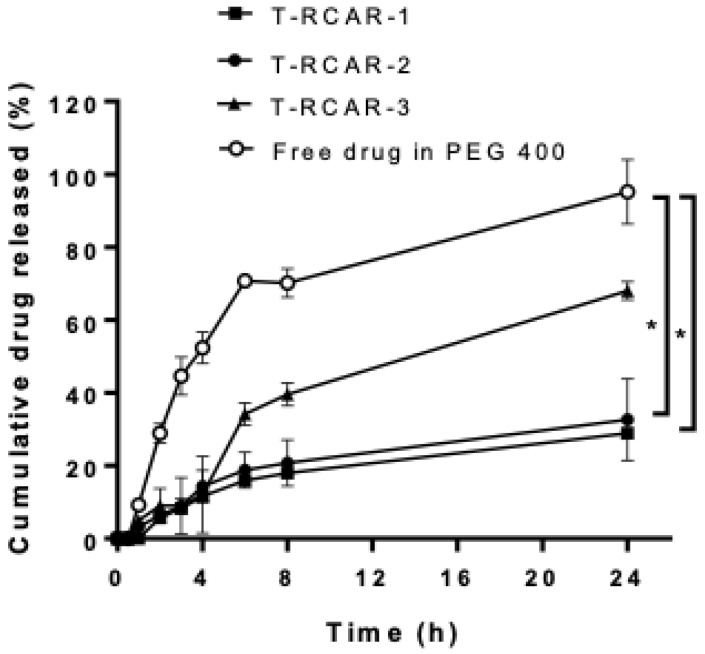
In vitro drug release profiles of free drug R-carvedilol dissolved in PEG 400 and three R-carvedilol-loaded transfersomes (T-RCARs). Formulations containing 100 μg of R-carvedilol (concentration 40 ug/mL in 2.5 mL PBS) were loaded in the Pur-A-Lyzer Maxi Dialysis tube, which was immersed in 100 mL of PBS (pH 7.4) as the release medium. Three independent samples were set up for each formulation. At various time points, 1 mL of samples were withdrawn from the medium and replaced with 1 mL of fresh PBS. The samples were analyzed via HPLC. Data are presented as mean ± SEM (n = 3). *: *p* < 0.05 based on ordinary one-way ANOVA and Dunnett’s multiple comparisons test.

**Figure 2 nanomaterials-13-00929-f002:**
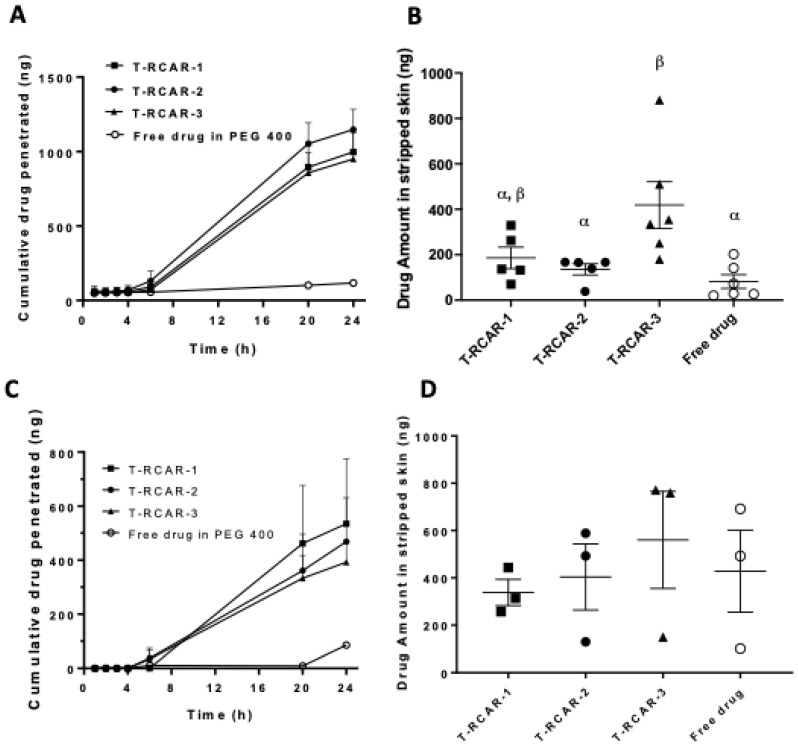
Ex vivo skin drug permeation profiles of T-RCARs using 40% PEG 400 in PBS or 4% BSA in PBS as receiver media. (**A**) Formulations containing 4 μg drug in 0.2 mL formulation solutions were loaded to the skin facing the donor compartments in the Franz diffusion cells. Data shown are cumulative R-carvedilol permeated into the receptor compartment as a function of time up to 24 h. The receiver compartment contains PBS containing 40% *v/v* of PEG400. Data are presented as mean ± SE (n = 5~6). (**B**) R-carvedilol levels in stripped skin (epidermal and dermal layers) 24 h after loading the drugs, determined via HPLC. Data are presented as mean ± SE (n = 5~6). (**C**) Cumulative R-carvedilol permeated into the receptor compartment as a function of time up to 24 h. The receiver compartment contains 4% BSA in PBS. Data are presented as mean ± SE (n = 3). (**D**) R-carvedilol levels in stripped skin (epidermal and dermal layers) 24 h after loading the drugs, determined via HPLC. Data are presented as mean ± SE (n = 3). An ANOVA followed by a Tukey–Kramer multiple comparisons test was used to assess statistical differences at *p* < 0.05, and differences denoted by different Greek letters.

**Figure 3 nanomaterials-13-00929-f003:**
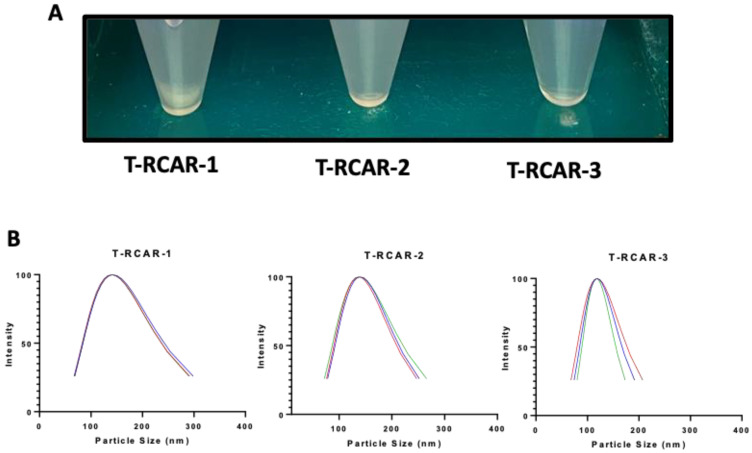
Transfersome visualization and size distribution after one year of storage in the refrigerator. (**A**) Photo of formulations after storage at 4 °C for one year. (**B**) Transfersome size distribution for the T-RCAR-1, -2, and -3 after storage for one year at 4 °C. The data plotted for each formulation are the triplicated reading for one sample (n = 1). Lines in different color: the particle size analysis was run three times for each formulation.

**Figure 4 nanomaterials-13-00929-f004:**
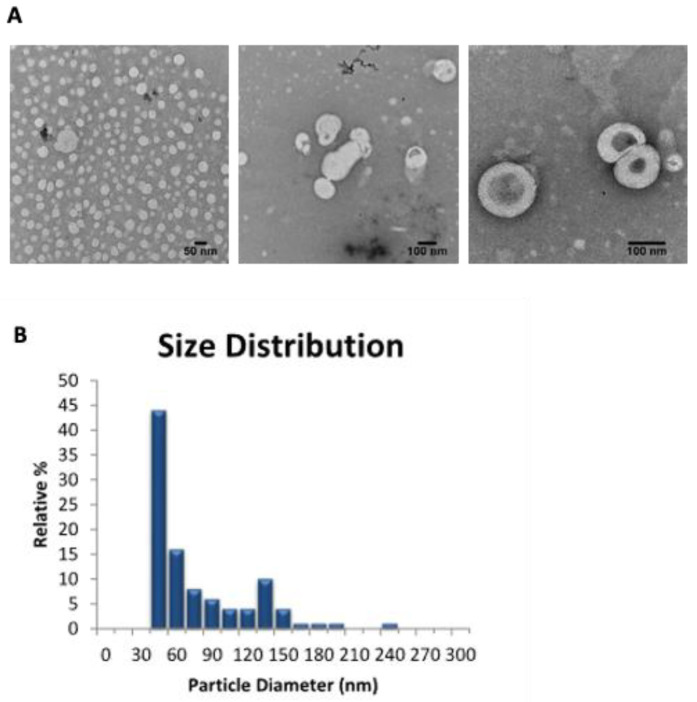
Transmission electron microscopy (TEM) analysis. (**A**) Representative TEM photomicrographs of T-RCAR-3. The scale bar represents 50 or 100 nm. (**B**) The particle size and distributions in frequency (%) for T-RCAR-3 derived from the TEM analysis. Only one batch of T-RCAR-3 (#T-RCAR-3-3) was analyzed by TEM (n = 1).

**Figure 5 nanomaterials-13-00929-f005:**
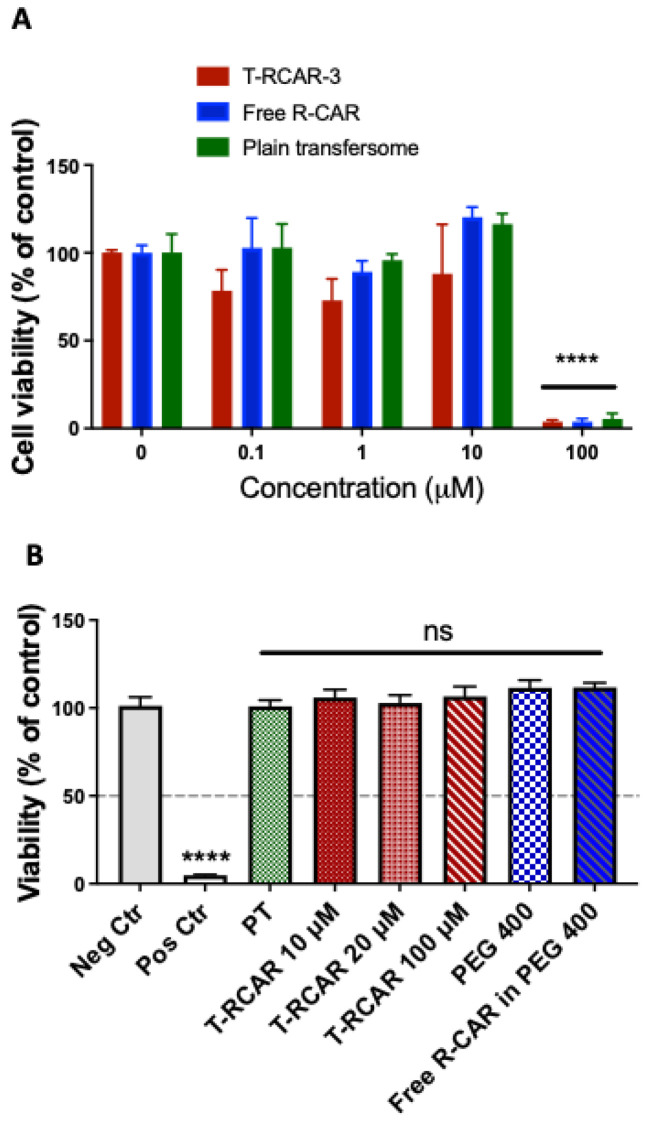
In vitro irritation test. (**A**) The 2D skin irritation test on JB6 cell culture. The cells were cultured in 96-well plates and treated with T-RCAR-3, free drug or PT for 48 h. MTT assay was used to evaluate the cell viability. The data were normalized by controls without treatment (n = 3). Only in treatment at the highest concentrations was there a statistically significant reduction in cell viability (ANOVA). ****: *p* < 0.0001. (**B**) The 3D skin irritation test on EpiDerm. MTT assay was conducted on EPI-200 tissues after topical applications of negative control (sterile DPBS), positive control (strong irritant, 5% SDS) or test agents in various doses for 60 min. The data were normalized by negative controls (n = 3). Only the positive controls (5% SDS) showed statistically significant reduction in cell viability (ANOVA). ****: *p* < 0.0001. ns: not significant.

**Figure 6 nanomaterials-13-00929-f006:**
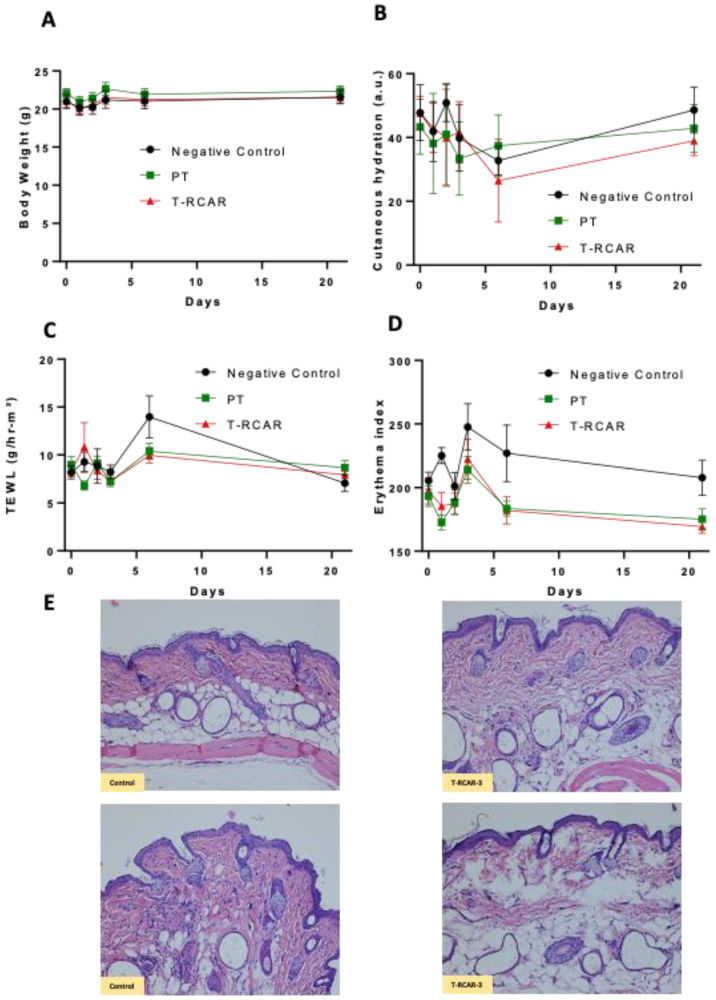
In vivo dermal toxicity test (repeated dose). (**A**) Body weight of mice was measured one day before, and on days 1, 2, 3, 6, and 21 after daily T-RCAR gel topical treatment (100 μM R-carvedilol). (**B**) Cutaneous hydration levels as measured with the corneometer one day before, and days 1, 2, 3, 6, and 21. On day 6, T-RCAR showed reduced hydration in 4 out of 6 mice < 30, indicating skin dryness, but recovered at 3 weeks. (**C**) Transepidermal water loss (TEWL) for skin barrier function, measured by the Tewameter one day before, and days 1, 2, 3, 6, and 3 weeks after T-RCAR topical treatment was initiated; TEWL < 25: normal condition. (**D**) Mexameter was used to measure erythema, one day before, and days 1, 2, 3, 6, and 3 weeks after T-RCAR topical treatment was initiated; <330: minimal erythema. Sample size: n = 4 for control; and n = 6 for PT or T-RCAR. Statistical analysis was based on repeated measures ANOVA. For most parameters, there was no significant difference between the three groups, except the erythema index, where the control group showed a slightly increased erythema. (**E**) Representative H&E-stained images for skin tissues collected from control mice and T-RCAR-3-treated mice.

**Figure 7 nanomaterials-13-00929-f007:**
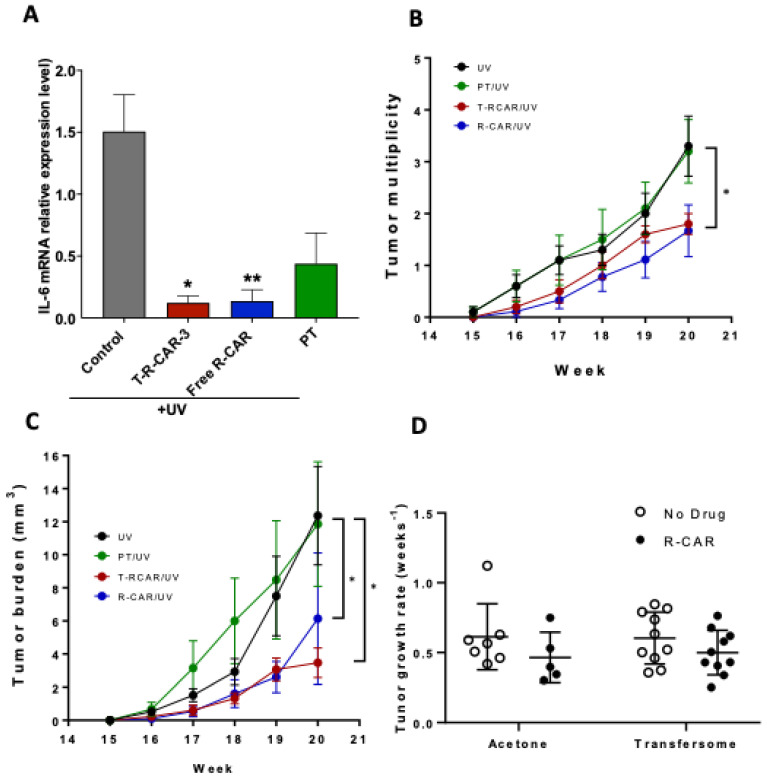
Effects of T-RCAR-3 and free R-carvedilol on UV-induced skin inflammation and tumor formation in SKH-1 mice. (**A**) Mice were pre-treated with vehicle (PEG 400), T-RCAR-3 gel, free drug in PEG 400 or plain transfersome (PT) gel daily for two days, exposed to single dose UV (336 mJ/cm^2^), followed by a third dose of topical treatment. The skin tissues were collected six hours after UV exposure for RNA isolation. The mRNA expression of IL-6 was examined via qRT-PCR analysis (n = 3~5). An ANOVA followed by a Tukey–Kramer multiple-comparison post hoc test was used to assess statistical differences at *p* < 0.05. All data are shown with the mean ± SE. *: *p* < 0.05; **: *p* < 0.01. (**B**) Mice were pre-treated with T-RCAR-3 or R-CAR in acetone for two weeks. The mice were then exposed to gradual doses of UV up to 150 mJ/cm^2^ three times per week, and drug treatments were given immediately after each irradiation. The number of tumors per mouse and average tumor volume per mouse (**C**) are plotted and analyzed via a chi-square analysis to assess statistical differences at *p* < 0.05 (*) (n = 10). (**D**) Tumor growth rates calculated by fitting the tumor volume data to an exponential growth formula and solving for the rate. An ANOVA was run on the rates, *p* = 0.366593.

**Figure 8 nanomaterials-13-00929-f008:**
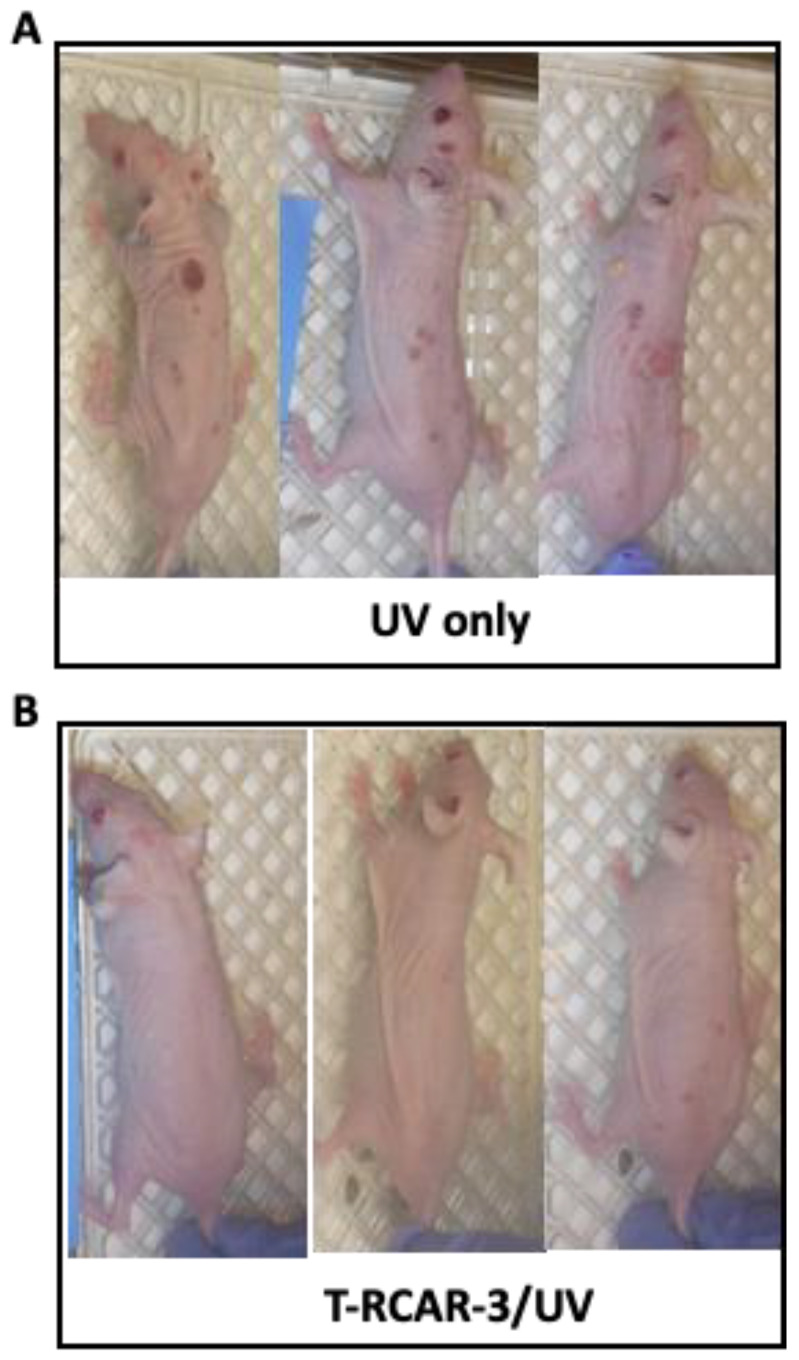
Effects of T-RCAR-3 on UV-induced tumor formation in SKH-1 mice. Mice were pre-treated with T-RCAR-3 for two weeks and were exposed to gradual doses of UV up to 150 mJ/cm^2^ three times per week; T-RCAR-3 treatments were given immediately after each irradiation. Representative photographs of mice from the UV-treated and UV-treated plus T-RCAR-3 at week 20 are shown. (**A**) Mice that were exposed to UV, without drug treatment. (**B**) Mice that were treated by T-RCAR-3 and exposed to UV.

**Table 1 nanomaterials-13-00929-t001:** Physical characteristics of R-carvedilol-loaded transfersomes prepared with Tween 80 or sodium cholate and comparison of the ex vivo skin penetration.

ID *	Ratio			Characterization				24-h Skin Penetration (%)
	Drug	SPC	Edge Activator	Mean Particle Size (nm) ± SD	PDI ± SD	Zeta Potential (mV) ± SD	% Encapsulation Efficiency	
F1S	1	3	0	106.15 ± 2.98	0.15 ± 0.02	8.0 ± 0.67	91.6	27
F2S	1	3	0.25	116.73 ± 1.19	0.22 ± 0.01	16.5 ± 4.72	96.5	22
F3S	1	3	0.5	96.50 ± 2.73	0.15 ± 0.01	8.0 ± 0.98	83.6	23
F1T	1	3	0	110.60 ± 0.57	0.16 ± 0.02	15.4 ± 0.60	71.5	25
F2T	1	3	0.25	107.02 ± 1.36	0.16 ± 0.02	10.34 ± 0.74	82.7	34
F3T	1	3	0.5	100.36 ± 0.39	0.17 ± 0.02	8.16 ± 2.06	80.6	33

*: S: sodium cholate; T: Tween 80.

**Table 2 nanomaterials-13-00929-t002:** Physical characteristics of R-carvedilol-loaded transfersomes.

Batch ID	Formulation ID	Ratio			Characterization			
		Drug	SPC	Tween 80	Mean Particle Size (nm) ± SD	PDI ± SD	Zeta Potential (mV) ± SD	% Encapsulation Efficiency
1-1	T-RCAR 1	1	3	0	110.6 ± 0.57	0.18 ± 0.02	34.4 ± 0.60	71%
1-2	T-RCAR 1	1	3	0	91.88 ± 0.95	0.17 ± 0.02	38.83 ± 1.45	92%
1-3	T-RCAR 1	1	3	0	119.77 ± 1.75	0.13 ± 0.03	37.53 ± 1.23	71%
					107.42 ± 14.21	0.16 ± 0.026	36.92 ± 2.28	78 ± 12
2-1	T-RCAR 2	1	3	0.25	107.02 ± 1.36	0.16 ± 0.02	30.34 ± 0.74	83%
2-2	T-RCAR 2	1	3	0.25	91.46 ± 1.51	0.16 ± 0.05	32.87 ± 1.67	91%
2-3	T-RCAR 2	1	3	0.25	123.30 ± 4.94	0.12 ± 0.02	33.67 ± 1.89	81%
					107.26 ± 15.92	0.15 ± 0.023	32.29 ± 1.74	85 ± 5.5
3-1	T-RCAR 3	1	3	0.5	100.36 ± 0.39	0.17 ± 0.02	32.16 ± 2.06	81%
3-2	T-RCAR 3	1	3	0.5	89.22 ± 1.37	0.16 ± 0.02	34.6 ± 0.46	89%
3-3	T-RCAR 3	1	3	0.5	101.52 ± 3.85	0.14 ± 0.02	37.8 ± 0.15	81%
					97.03 ± 6.79	0.16 ± 0.015	34.85 ± 2.83	84 ± 4.6

**Table 3 nanomaterials-13-00929-t003:** Stability data for T-RCAR formulation 3 (T-RCAR-3).

Week	Mean Particle Size (nm) ± SD	PDI ± SD	% Encapsulation Efficiency
T-RCAR formulation 3 batch 1			
1	105.11 ± 2.75	0.076 ± 0.020	83
2	114.97 ± 3.32	0.06 ± 0.02	
3	108.98 ± 0.77	0.07 ± 0.01	
4	113.59 ± 3.26	0.06 ± 0.02	
5	111.52 ± 0.7	0.06 ± 0.01	83
T-RCAR formulation 3 batch 2			
1	96.01 ± 3.75	0.07 ± 0.011	81
2	106.73 ± 2.68	0.07 ± 0.01	
3	102.28 ± 0.57	0.05 ± 0.02	
4	105.24 ± 2.21	0.09 ± 0.01	
5	104.36 ± 1.75	0.08 ± 0.01	83
T-RCAR formulation 3 batch 3			
1	108.93 ± 2.73	0.096 ± 0.027	83
2	118.34 ± 1.32	0.08 ± 0.03	
3	115.8 ± 0.6	0.07 ± 0.01	
4	119.99 ± 3.16	0.07 ± 0.04	
5	116.72 ± 0.45	0.07 ± 0.03	83

## Data Availability

Data is contained within the article.

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
