# Peer review of "Transfersome Encapsulated with the R-carvedilol Enantiomer for Skin Cancer Chemoprevention"

_nanomaterials, 2023, doi:10.3390/nano13050929_

Round 1

Reviewer 1 Report

This study provides preclinical evidence that R-carvedilol loaded transfersomes can effectively prevent UV-induced skin cancer without any major adverse effects. The research work is interesting and with potential translational values as a chemo-preventive agent to UV induced skin cancer. Howeverthe mechanisms of action by R-carvedilol loaded transfersomes are very limit in the article. UV induced skin cancer is known related to oxidative stress, inflammation,DNA damage and cell proliferation. The authors should provide more solid evidences include  (1) representative biomarkers, and (2) pathological characteristics of animal model both early and late stage after UV exposure; and to confirm the precise target how the transfersomes blocked UV damage to the skin.  

Author Response

We agree with the reviewer that this manuscript does not cover comprehensive mechanistic studies. R-carvedilol and the racemic carvedilol which contains R- and S-carvedilol attenuate UV radiation medicated skin lesions via multiple mechanisms of action. In our previously published work (Prevention of Skin Carcinogenesis by the Non-β-blocking R-carvedilol Enantiomer, Cancer Prev Res. 2021. PMID: 33648941), we conducted several experiments to determine the effects of R-carvedilol on UV-induced oxidative stress, inflammation, DNA damage, in vitro and in vivo.

The R-carvedilol we used in our previous study was in the free drug form, dissolved in acetone (as a solvent and a skin penetration enhancer, for mouse application). In the current study, our main purpose was to explore the potential of transfersome as a delivery vehicle for enhancing R-carvedilol’s skin permeability. Such transfersomal formulation for R-carvedilol is more likely used for clinical development, since acetone is just an experimental solvent to prove the concept.

Since the major goal is to develop and characterize a drug delivery method, we did not repeat all mechanistic assays, but selected a few, for example, using IL-6 expression as the representative pro-inflammatory biomarker to examine T-RCAR’s efficacy. Furthermore, since pathological characteristics of UV radiation of mice have been reported in numerous publications, and because established UV exposure protocols were used, pathological mechanisms for these models are not fully described in this manuscript.

For the precise target for R-carvedilol, studies in our lab are on-going. We revised the “Discussion” (page19) to address reviewer’s comments.

Reviewer 2 Report

Dear Authors,

The manuscript tilted “Transfersome encapsulated with the R-carvedilol enantiomer for skin cancer chemoprevention” is well written and meaningful work. Some comments below.

1.       In general, the introduction provides little new information and can be improved. Recent literature from the field can be added. Newer and more comprehensive literature can be cited here and throughout the introduction. Suggestions - https://doi.org/10.1016/j.addr.2021.113929 , https://doi.org/10.1016/j.addr.2022.114293

2.       Transferosomes used in this study were flexible? Can you please add tests that demonstrated this?

3.       Table 1 shows 24-hr skin penetration data, is this historical or new data?

4.       The animal skin models used are fine but their barrier properties are unlike human skin. Excised human skin where available should be used. Can the authors please comment.

5.       Similarly, irritancy test were performed on 3D skin models which are shown to have inferior barrier properties. Can the authors include comments on this.

6.       The impact of formulation on hydration of the skin has been explored. Corneometer readings demonstrated variability but no significant changes. Previous literature (https://doi.org/10.1208/s12248-016-9984-0) demonstrates microscopic methods to assess hydration increase. This could have been performed on animal skin.

7.       Figure 7 E can be separated, and larger images would be more representative.

Author Response

Reviewer #2:

The manuscript tilted “Transfersome encapsulated with the R-carvedilol enantiomer for skin cancer chemoprevention” is well written and meaningful work. Some comments below. 

  1. In general, the introduction provides little new information and can be improved. Recent literature from the field can be added. Newer and more comprehensive literature can be cited here and throughout the introduction. Suggestions - https://doi.org/10.1016/j.addr.2021.113929, https://doi.org/10.1016/j.addr.2022.114293

To improve the “Introduction”, we read suggested review articles and cited relevant information in the manuscript and added more references throughout the manuscript, in the Introduction, Results and Discussion.

  1. Transferosomes used in this study were flexible? Can you please add tests that demonstrated this?

It has been reported that transfersomes are sufficiently flexible to pass through the pore with diameter much smaller than their own size. Although we haven’t tested the flexibility (we should do this in the future), the ex vivo skin permeation data in Fig.2 indirectly proves its flexibility.

  1. Table 1 shows 24-hr skin penetration data, is this historical or new data?

These are new data, not historical data, which have not been published previously.

  1. The animal skin models used are fine but their barrier properties are unlike human skin. Excised human skin where available should be used. Can the authors please comment.

We agree with the reviewer. This statement “Future studies should confirm this finding on excised human skin” has been added to the Discussion (page 19).

  1. Similarly, irritancy test were performed on 3D skin models which are shown to have inferior barrier properties. Can the authors include comments on this.

The limitation for the 3D skin model has been discussed and added into the Discussion (page 10).

  1. The impact of formulation on hydration of the skin has been explored. Corneometer readings demonstrated variability but no significant changes. Previous literature (https://doi.org/10.1208/s12248-016-9984-0) demonstrates microscopic methods to assess hydration increase. This could have been performed on animal skin.

We agree with the reviewers that although Corneometer is the most popular non-invasive tool to determine skin hydration, the safety data should be confirmed by other methods. The requested information has been added to the Results (page 14).

  1. Figure 7 E can be separated, and larger images would be more representative.

 Revision has been made to Fig.7 and Fig.8 was added, according to reviewer’s comments.

Reviewer 3 Report

Dear Authors

This paper is a very interesting work, well structured and clear presented.

 I have some remarks:

 1. Please explain the difference between zeta potential values recorded for batches T-RCAR 1 – T-RCAR 3 (Table 2) and for F1T – F3T (Table 1), respectively.

2. Concerning the Sections 3.6-3.8, I do not have the expertize to juge the results obtained.

3. Minor English revision and spelling corrections are required.

Author Response

This paper is a very interesting work, well structured and clear presented.

I have some remarks:

  1. Please explain the difference between zeta potential values recorded for batches T-RCAR 1 – TRCAR 3 (Table 2) and for F1T – F3T (Table 1), respectively.

We have added a brief explanation for such discrepancy into the Results (page 7). One possibility is that when we prepared the pilot formulations in Table 1 we sonicated the suspension for only 5 min. Later, when we repeated the formulation preparation to make additional batches of transfersomes we sonicated the formulations for 30 min, because we found that longer sonication avoided the formation of clusters and increased the stability.

  1. Concerning the Sections 3.6-3.8, I do not have the expertize to juge the results obtained.

We have revised the manuscript according to all the four reviewers’ suggestions. We are grateful that the reviewers come from different areas of expertise.

  1. Minor English revision and spelling corrections are required.

We slightly modified text in the wording after careful grammar checking (see highlights by “track change” in the text file).

Reviewer 4 Report

The manuscript is about a very interesting study on a novel possibility to transfer carvedilol. A few aspects are important in order to increase the value of the paper:

(1) were the reagents purified before their use?

(2) solvent was evaporated under reduced pressure; low, medium, high or ultra-high vacuum?

(3) add a briefly description of encapsulation efficiency (formula, procedure), not just a reference

(4) add the brand and model of TEM instrument and the conditions that were used in characterisation

(5) verify the humidity of mice environments - standards specify that "most animals do well at 40 to 60%"; does your low value affect the skin parameters, especially cutaneous hydration?

(6) you did not mention that the low values of Zeta potentials from Table 1 indicate a high tendency to form particles clusters; however, this aspect is not confirmed by the Mean particle Size. Can you explain why?

Author Response

The manuscript is about a very interesting study on a novel possibility to transfer carvedilol. A few aspects are important in order to increase the value of the paper:

(1) were the reagents purified before their use?

R-carvedilol used in our studies was synthesized at Chem-Impex International Inc. (Wood Dale, IL, USA). The purity was determined using Chiral HPLC as 98.97% at the company. After the compound was received, the accuracy and purity were confirmed in our lab by chiral HPLC using Phenomenex Lux® 5 µm Cellulose-4 LC Column 250 x 4.6mm (Phenomenex, Torrance, CA, USA). The information has been added to the Materials and Methods (page 2).

(2) solvent was evaporated under reduced pressure; low, medium, high or ultra-high vacuum?

The solvent was evaporated under the pressure of 100 mbar (~ 20 min), which is considered medium pressure.

(3) add a briefly description of encapsulation efficiency (formula, procedure), not just a reference

The detailed approach has been added to the Materials and Methods (page 3).

(4) add the brand and model of TEM instrument and the conditions that were used in characterisation

The detailed methods for TEM analysis has been added to the Materials and Methods (page 4).

(5) verify the humidity of mice environments - standards specify that "most animals do well at 40 to 60%"; does your low value affect the skin parameters, especially cutaneous hydration?

Our animal house has humidity 35%. The environment could affect skin hydration. Our data should be verified using other approach. This statement has been added to the Results (page 14).

(6) you did not mention that the low values of Zeta potentials from Table 1 indicate a high tendency to form particles clusters; however, this aspect is not confirmed by the Mean particle Size. Can you explain why?

One possibility is that when we prepared the pilot formulations in Table 1 we sonicated the suspension for only 5 min. Later, when we repeated the formulation preparation to make additional batches, we sonicated the formulations for 30 min, because we found that longer sonication avoided the formation of clusters and increased the stability. This statement has been added into the Results (page 7).

Round 2

Reviewer 1 Report

The authors have answered my concers in the revised version.